# Improving equity in access to early diagnosis of cancer in different healthcare systems of Latin America: protocol for the EquityCancer-LA implementation-effectiveness hybrid study

Maria-Luisa Vázquez ![ORCID],[1] Ingrid Vargas ![ORCID],[1] Maria Rubio-Valera,[2] Ignacio Aznar-Lou,[3] Pamela Eguiguren,[4] Amparo-Susana Mogollón-Pérez,[5] Ana-Lucía Torres,[6] Andrés Peralta ![ORCID],[6] Sónia Dias,[7] Signe Smith Jervelund,[8] For EquityCancer-LA

For numbered affiliations see end of article.

**Correspondence to**
Dr Ingrid Vargas;
ivargas@consorci.org

## ABSTRACT

**Introduction** Healthcare fragmentation, a main cause for delay in cancer diagnosis and treatment, contributes to high mortality in Latin America (LA), particularly among disadvantaged populations. This research focuses on integrated care interventions, which have been limitedly implemented in the region. The objective is to evaluate the contextual effectiveness of scaling-up an integrated care intervention to improve early diagnosis of frequent cancers in healthcare networks of Chile, Colombia and Ecuador.

**Methods and analysis** This research is two pronged: (A) quasi-experimental design (controlled before and after) with an intervention and a control healthcare network in each LA country, using an implementation-effectiveness hybrid approach to assess the intervention process, effectiveness and costs; and (B) case study design to analyse access to diagnosis of most frequent cancers. Focusing on the most vulnerable socioeconomic population, it develops in four phases: (1) analysis of delays and barriers to early diagnosis (baseline); (2) intervention adaptation and implementation (primary care training, fast-track referral pathway and patient information); (3) intracountry evaluation of intervention and (4) cross-country analysis. Baseline and evaluation studies adopt mixed-methods qualitative (semistructured individual interviews) and quantitative (patient questionnaire survey) methods. For the latter, a sample size of 174 patients with cancer diagnosis per healthcare network and year was calculated to detect a proportions difference of 15%, before and after intervention ($\alpha$=0.05; $\beta$=0.2) in a two-sided test. A participatory approach will be used to tailor the intervention to each context, led by a local steering committee (professionals, managers, policy makers, patients and researchers).

**Ethics and dissemination** This study complies with international and national legal stipulations on ethics. It was approved by each country's ethical committee and informed consent will be obtained from participants. Besides the coproduction of knowledge with key stakeholders, it will be disseminated through strategies such as policy briefs, workshops, e-tools and scientific papers.

## STRENGTHS AND LIMITATIONS OF THIS STUDY

⇒ The study uses implementation-effectiveness hybrid research to generate evidence in Latin America on the effectiveness, implementation process and cost of scaling-up an integrated care intervention to improve early diagnosis of cancer.

⇒ It will use comprehensive national and cross-national analysis of time intervals, key barriers and facilitators of access to early cancer diagnosis of the most frequent cancers in Latin America countries, adopting a common theoretical framework and mixed-method approach.

⇒ The adoption of a participatory approach, involving key stakeholders, will facilitate the adaptation of the intervention to local needs and its uptake, and contribute to overcome contextual barriers and increase sustainability.

⇒ A potential limitation might be the political and managerial changes during the intervention implementation process, leading to limited institutional support for the interventions and its participatory process.

⇒ The absence of a centralised and updated cancer registry in the study areas may hinder and delay the survey data collection.

## INTRODUCTION

Delays in diagnosis and treatment contribute to the high and steadily increasing cancer mortality rates in Latin America (LA).[1–4] WHO recently established early cancer diagnosis programmes as a priority for low-income

and middle-income countries (LMICs)[5–7] and defined it as the early identification of cancer in patients who have symptoms of the disease.[5] The objective of early cancer diagnosis—which is the focus of EquityCancer-LA—is to identify the disease in people who have symptoms and signs consistent with cancer at the earliest possible opportunity and to link it with treatment without delay.[5] Interventions to ensure adequate and timely access and strong care coordination across the required health services (primary care (PC), diagnostic and specialist services) are key for cancer control. Early diagnosis and referral have been affected by the COVID-19 outbreak responses (confinement, resources diversion and interruption of screening programmes, etc), making interventions in this area more relevant than ever.[8–10]

The patient pathway from the discovery of symptoms to a confirmed diagnosis of cancer is often very complicated, encompassing different healthcare levels, services and professionals and contributing to delays in treatment. While delays may differ according to patient and healthcare system characteristics, for its analysis (and to define interventions to address them) two types of intervals are generally distinguished[11]: the patient interval, or time taken from the discovery of symptoms to seeking a medical professional, and the provider interval, from the request of the first medical consultation to the confirmation of cancer diagnosis.

Delays in cancer diagnosis are accentuated in fragmented healthcare systems such as those in LA, characterised by limited access to health services and poor coordination of care among a wide range of providers. Although further research is required, available studies in LA suggest that most delays occur within the provider interval[4 12 13] and are related to financial, geographic and organisational barriers of access, including poor diagnostic suspicion in PC and limited cross-level care coordination.[3 14 15] Population groups most affected are those of low socioeconomic status living in large urban slums, rural inhabitants, indigenous populations[3] or those on the subsidised scheme in Colombia (no formal employment),[16] resulting in large inequities in early diagnosis.[2 17]

### Integrated care as a strategy to improve early cancer diagnosis

The implementation of integrated care interventions focusing on strengthening PC and improving care coordination across care levels are considered essential.[18–20] This is particularly so in PC-based health systems—as in most Latin American countries[21]—in which PC is the entry point to healthcare and where, around 85% of cancers are diagnosed after symptomatic presentation to PC,[18] including cancers for which screening programmes exists.[22] Therefore, to improve the early diagnosis of cancer, PC doctors should be able to assess the probability of cancer, and to refer rapidly suspected patients with cancer to specialised care for cancer investigations and treatment.[18 22] To this end, comprehensive multicomponent interventions, combining PC training with improved diagnostic and treatment pathways and patient information to navigate the health services, are increasingly recommended,[23] also for LMICs,[5–7] but introduced mainly in high-income countries (HICs).[22 24–28] In LA, current national cancer plans and strategies[29–34] promote integrated cancer care strategies to address fragmentation and geographic inequity in cancer diagnosis and care. However, the implementation of these integrated care strategies is very limited, their evaluation scarce[35–37] and focused on specific mechanisms.[38–41]

### The effectiveness of integrated care as a strategy to improve early cancer diagnosis

Available evidence, mostly from HICs, shows that the implementation of continuing education on early diagnosis of PC teams, fast-track referral pathways and guidelines and patient information to navigate the health services are effective[22 25 26 42–44] and efficient[22 45–47] in improving early diagnosis of cancer. Integrated care strategies can improve the appropriateness of referrals and tests and increase diagnostic accuracy for PC doctors,[22 42 43] which can in turn reduce the provider delay.[18 24 26] Moreover, the use of information and communication technology (ICT) in PC training (eg, videoconferencing) and virtual clinical consultation show multiple benefits (eg, increased access to secondary/tertiary care (SC/TC), care coordination across levels) especially in big cities with traffic problems and rural areas.[48–52]

Evidence also shows that interventions are only successful if adapted and implemented taking into account local problems (delays, pathways and barriers),[14] priorities and organisational factors and are agreed on by the professionals involved across care levels.[53–59] In this respect, participatory and multidisciplinary approaches, including participatory action research (PAR) has developed a growing reputation as an effective means for tailoring interventions and achieving organisational change,[60] although its application to improve early diagnosis of cancer has been rare.[61] PAR main characteristics are: non-linear and reflexive process, collaboration between local actors and researchers, involvement of the local stakeholders throughout the process, research teams as facilitators (capacity building, systematisation, monitoring and feedback), and based on three basic principles, egalitarian participation, shared decision-making and coproduction of knowledge.[62] Involving main actors lends greater contextual relevance and validity to interventions, creates more interest and increases sustainability.[62 63] Moreover, participatory training methods improve not only knowledge and skills but also favour coordination between professionals and stakeholders throughout the healthcare network.[64]

### The need for implementation research in LA

Implementation research is needed to assess the extent to which an effective intervention for early diagnosis of cancer is also effective in a specific health system and social context. This approach also helps to determine the

factors of success or failure in the intervention implementation related to the context and the process, including the keys to its sustainability, affordability and large scale use.[65][66] However, limited implementation research has been conducted in LMICs, including LA[64][67]—and even less to evaluate interventions for early diagnosis of cancer. Results of the Equity-LA II implementation research in six LA countries—one of the few carried out in the region so far—highlighted on the one hand, the influence of contextual factors in the implementation of integrated care interventions such as, interference of the political cycle, limited interest and time of professionals. On the other hand, it showed how the participatory approach applied in designing and implementing the interventions enhanced professional motivation to adopt them and continuous institutional support, which ultimately contributed to modifying some contextual barriers.[46]

### Aim of the EquityCancer-LA project

The aim of EquityCancer-LA is to evaluate the contextual effectiveness of scaling-up a multi-component integrated care intervention to improve early diagnosis of cancer, taking a participatory approach, in healthcare networks of different healthcare systems in LA. This project builds on frameworks, results and best practices in implementing integrated care interventions using a PAR approach, generated by Equity-LA II[64] research project. This paper outlines the study protocol of EquityCancer-LA.

## METHODS AND ANALYSIS
### Study design

EquityCancer-LA adopts a double design: (A) a quasi-experimental design (a controlled before-and-after design) with an intervention and a control health services network in each LA country, using a participatory and implementation-effectiveness hybrid approach[65] to assess the intervention implementation process, contextual effectiveness—that is, under real-life conditions—and costs, and (B) a case study design,[68] using mixed-methods,[69] for the in-depth analysis of key barriers and facilitators to early cancer diagnosis of most frequent cancers and its implications for equity of access.

### Analytical frameworks adapted for the evaluation of the intervention effectiveness and analysis of access to early diagnosis of cancer

This study is underpinned by two conceptual frameworks:

(1) Analytical framework for the evaluation of an intervention effectiveness and implementation process (figure 1), based on the Equity-LA II developed framework.[64] It includes (A) the evaluation of the intervention effectiveness and cost-effectiveness based on intermediate and final outcomes related to cancer diagnosis and (B) the analysis of the intervention process and how it affects its effectiveness, embracing the three dimensions of Pettigrew and Whipp[66][70]: the context, the process of design and implementation, and the content of the

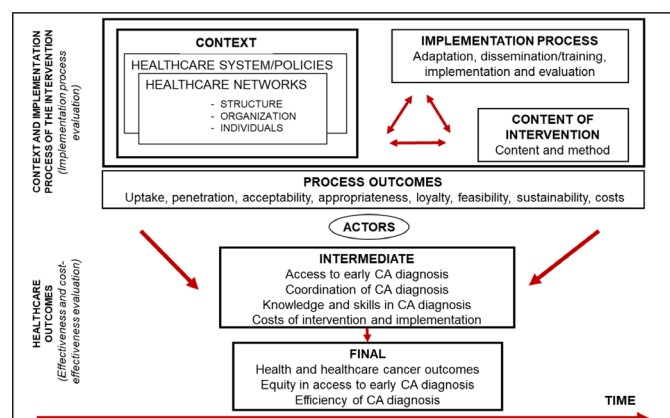

**Figure 1** EquityCancer-LA conceptual framework for comprehensive evaluation of the intervention to improve early diagnosis of CA adapted from Equity-LA II.[64] CA, cancer; LA, Latin America.

intervention. Context and process elements interact in a complex way over time influencing, in the first place, the results of the process of the intervention implementation (process outcomes) and ultimately, its intermediate and final outcomes. The most commonly used process outcomes, are those defined by Proctor et al[71]: adoption or uptake, penetration, acceptability, appropriateness, feasibility, intervention costs and sustainability. The effectiveness analysis establishes any significant changes in the outcomes that can be attributed to the integrated care intervention to improve early diagnosis of cancer. Intermediate outcomes refer to patient access to early diagnosis of cancer, to cross-levels care coordination for cancer diagnosis, knowledge of risk factors, associated symptoms and diagnostic ability of participant PC doctors; and the costs of cancer early diagnosis. Final outcomes refer to healthcare and health outcomes (eg, diagnostic delay; diagnostic stage of cancer), equity and efficiency in early diagnosis and care. The analysis of both contextual factors and the implementation results and costs are carried out taking into account key actors (health professionals, managers and users) viewpoint. The evaluation of the costs of the implementation, as part of the analysis of the intervention process, will be oriented by the stages of implementation completion (SIC) framework.[72]

(2) Framework for the analysis of access to early diagnosis of cancer, built on the model for the analysis of pathways to cancer diagnosis and treatment by Walter et al,[7311] and incorporating factors influencing healthcare access, as proposed by Aday and Andersen[74–76] (online supplemental figure 1). According to this framework, delays throughout the patient trajectory to confirmation of cancer diagnosis and initiation of treatment can be influenced by characteristics of the population at risk: need, predisposing (sociodemographic, believes, attitudes, knowledge) and enabling (income, health insurance, residence, etc) factors, as well as health services' factors: resources availability and organisation of health services (fee-for-service, waiting times, schedules, referral system,

etc).[77] Health policies can in turn affect access barriers related to the health services or changing (mutable) characteristics of the population at risk.[77]

## Study context

The field work of EquityCancer-LA is being carried out in three LA countries, Chile, Colombia and Ecuador, and within them, in two public health services networks providing care to middle-low and low-income areas. Although the study countries have different health system models, all are segmented by population groups according to socioeconomic or employment status,[78 79] with a public subsystem and a private one. The public sector, which is the focus of this study, is financed by social security contributions and/or taxes. It encompasses at least one subsystem dependent on the ministry of health, which is decentralised to different levels of government (departments/provinces and/or municipalities) and is generally aimed at the lower-income population and/or those without social security. In each country, two comparable health services networks (one intervention (IN) and one control (CN)) have been selected according to the following criteria: (A) provision of a continuum of services including at least PC and SC/TC, responsible for diagnosis and treatment of cancer; (B) provision of services to a defined population; (C) provision of care mainly to urban low and middle low socioeconomic areas; (D) willingness to participate and implement the intervention and (E) leadership with competence for implementing the intervention. Selected networks are: in Chile, the northern network (IN) and southern network (CN) of Santiago; in Colombia, central and northern health regions (IN) and southern health region (CN) of Cundinamarca; in Ecuador, 17D06 (IN) and 17D03 (CN) districts in the southern and northern areas of Quito.

## Research phases and methods

EquityCancer-LA is structured in four research phases which extends over 60 months (May 2021–April 2026), taking into account Campbell *et al* model for the evaluation of complex interventions,[80] and keeping with its

subsequent update,[81] and adopts an interdisciplinary, participatory and mix-methods approach (figure 2). The phases are: (I) an analysis of key barriers and facilitators for early diagnosis (baseline study), using both qualitative and quantitative methods, which is currently underway; (II) participatory adaptation and implementation of the intervention; (III) monitoring and evaluation of the intervention and a (IV) cross-country comparative analysis. At the beginning of the project, a local steering committee (LSC) of relevant stakeholders involved in the early diagnosis of cancer in the intervention network was established in each country. It is made up of health professionals from different care levels, managers, healthcare insurers (in Colombia), policy makers nominated by the participating institutions, as well as patients and the research team. The LSC will remain active throughout the project, and will be involved in all research phases, being in charge of the adaptation and implementation of the intervention and dissemination of results.

I. Analysis of key barriers and facilitators of early cancer diagnosis of the most frequent cancers (baseline) and implications for equity of access (32 months). During the first 6 months of this phase, the theoretical framework and research plan were finalised, and on the basis of available data, the LSCs selected frequent cancers to be addressed: breast, cervical, colorectal, stomach, lung, prostate, kidney and bladder cancers. The next step is to analyse the access to diagnosis of cancer in the study networks. This analysis employs a mixed-methods approach, combining qualitative and quantitative research methods of social sciences for data collection in an iterative and complementary way.

1. A qualitative study to analyse barriers and facilitators of access and patient's pathways to early diagnosis in each particular context from the stakeholders' perspective (19 months). A criterion sample was designed to include all discursive variants on the phenomenon: health professionals, administrative personnel, managers, policy makers and patients with cancer, with particular focus on vulnerable groups (low socioeconomic status, gender, ethnicity, distance to SC/TC services and rurality). Patients are selected from the databases of patients diagnosed with cancer of the health services (in Colombia also from the insurers), and other informants from a list of possible candidates according to the selection criteria, provided by an institutional contact. The final sample size will be reached by saturation of information. Data collection takes place in two phases, before and after the patients' questionnaire survey, using semistructured individual interviews and policy documents' analysis. Focus groups with health professionals to explore some issues in depth, might be conducted, if considered necessary and feasible in the pandemic context. Topic guides were developed with one common and one specific section for each informant group. A thematic analysis is being conducted with support of a specialised software (Atlas-ti, NVivo) to identify main contextual barriers and facilitators of cancer diagnosis to inform the adaptation

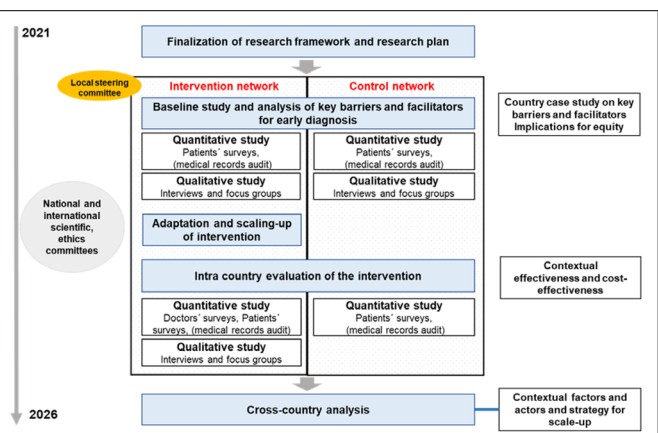

**Figure 2** EquityCancer-LA participatory, mix-methods approach and research design. LA, Latin America.

of the intervention in each area. In the second phase, critical cases identified by the patient survey might be invited to become case studies for an in-depth analysis to better capture patient journey complexity, reasons for delays in the diagnostic process as well as health consequences.

2. A quantitative study (9 months). A cross-sectional study by means of a questionnaire survey of patients diagnosed of cancer will be conducted in each country to establish differences in use of health services and pathways in cancer diagnosis, time intervals, barriers to early diagnosis among different population groups, costs and where possible, complemented with PC and SC/TC medical records audits. It will provide also the baseline for the evaluation of the intervention, to compare both networks before and after, and also the intervention and control networks. The study population will comprise patients aged 18 years or more with a confirmed cancer diagnosed and staged in the 3–6 months prior to the survey in the study healthcare networks. Patients with benign tumour, uncertain, with consecutive cancers in same organ (recurrence) or synchronous primary cancers will be excluded. The sample size of the study population has been estimated taking into account the controlled before and after design of the study and in order to ensure the detection of a 15% variation in time intervals, health services use and barriers to the early diagnosis of cancer in a bilateral contrast (before and after, intervention and control). A sample size of 174 patients with a diagnosis of cancer per healthcare network and year has been estimated, on the basis of a power of 80% (β=0.20) and a confidence level of 95% (α=0.05). Patients are selected from the databases of patients diagnosed with cancer of the health services (and/or insurers in Colombia). The results will contribute to adapt the intervention to the context, to analyse costs, barriers and inequalities in access to early diagnosis of cancer. The survey started in August 2022.

II. Participatory adaptation and implementation of the intervention to improve early diagnosis of frequent cancers in the countries of study, under real-life conditions (26 months).

EquityCancer-LA will adapt, scale-up and assess the feasibility of an effective, innovative and comprehensive integrated care intervention to reduce the provider delay(interval) in the diagnosis of the most frequent cancers. The components of the intervention are: (A) PC doctor's training to improve PC early diagnosis capacity and care coordination across care levels. According to the context, it will be on-line (videoconferencing) or face to face and reflexive and participatory methods will be used; (B) a fast-track referral pathway to expedite referral for diagnostic evaluation of patients jointly designed by PC and SC doctors, supported by the description of alarm symptoms that may raise cancer suspicion, standardised pathways or circuits for the diagnostic process and initial treatment, and referral guidelines agreed on by PC and

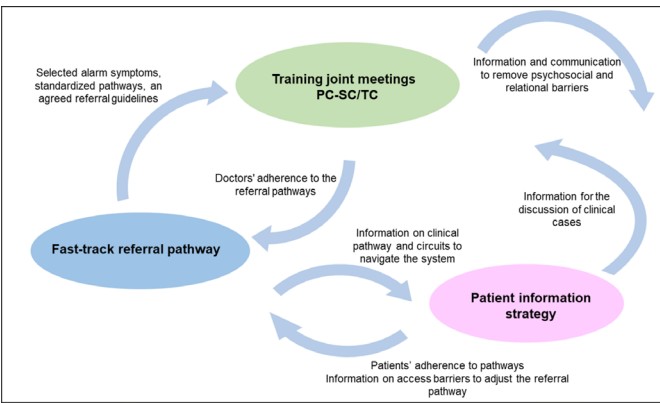

**Figure 3** The multicomponent integrated care intervention to improve early diagnosis of selected frequent cancers. EquityCancer-LA. LA, Latin America; PC, primary care; SC, secondary care; TC, tertiary care.

SC levels' doctors; and (C) a patient information strategy to assist the patients to navigate the system. The three components interact reinforcing each other effectiveness (figure 3). The intervention has been conceived based on available evidence, WHO recommendations for LMICs[5] and national cancer plans.[29–31]

A PAR approach will be used to tailor the multicomponent integrated care intervention to each context and implement it based on the preliminary results of the baseline study and the assessment of the implementation conditions, involving main stakeholders and building on previous Equity-LA II[64 82] experience and methods. A carefully phased approach will be used (figure 4). The participatory process starts in each LA country with the creation of the LSC. First, the evidence produced on the barriers and facilitators of early diagnosis of cancer and previous experiences related to the intervention in the healthcare networks will be discussed with the LSC and the interlevel group of professionals (GP), which will be created with those interested in taking action. The conditions for the intervention implementation (acceptability, appropriateness, feasibility, costs) and contextual barriers and facilitators will be evaluated with relevant stakeholders to introduce any necessary changes prior

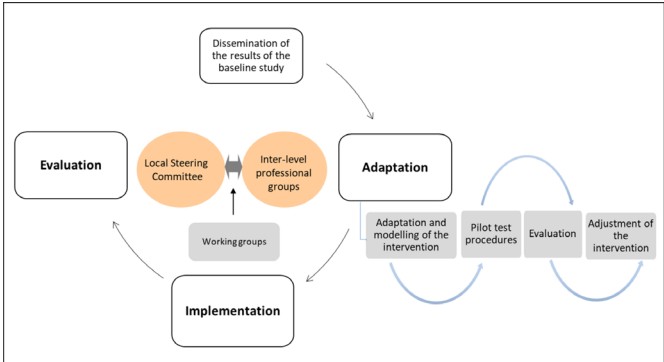

**Figure 4** The process of adapting and implementing the intervention to improve early diagnosis of cancer. EquityCancer-LA. LA, Latin America.

to its implementation. In Chile, an evaluation will be conducted of the conditions for the extension to the whole network of the joint virtual clinical conferences that were participatory designed and implemented in the framework of the Equity-LA II project.

In the second phase, the different components of the intervention will be adapted by the GP and LSC (contents, method, implementation plan and monitoring indicators) and some of them will be pilot-tested, if relevant. Interlevels working groups might be established for the adaptation of specific components, made up of participants from the different care levels and departments involved in the cancer care process. In this phase, the definitive plan for the intervention evaluation will be elaborated (including feasible and valid measures of intermediate and final outcomes, etc).

III. Monitoring and evaluation of the intervention will address both, the implementation process, as well as, the effectiveness under real-life conditions including the economic evaluation (cost-effectiveness and implementation costs), in each country to determine its sustainability and applicability to other contexts (35 months). Monitoring the intervention during the implementation process will facilitate its adaptation, the identification and correction of any contextual factors that may influence its implementation and the costs analysis. Indicators will be elaborated to monitor the implementation process (uptake, costs, etc) and to identify trends in effectiveness regarding intermediate and, if possible, final outcomes.

The evaluation of the implementation process will identify the factors related to the context, process and intervention content that have influenced the implementation and its effectiveness, as well as its sustainability and replicability to other contexts. It will be based mainly on a qualitative study, which will be essential to discriminate what component of the intervention was responsible for what results. Individual interviews and focus groups will be conducted with different stakeholders involved in the process. These are: LSC, interlevel GPs, managers, policy makers and patients. This study will be complemented with quantitative analyses of specific aspects: uptake, acceptability, appropriateness regarding specific problems in cancer early diagnosis (process indicators; doctor survey).

The evaluation of the effectiveness will encompass two intra-country comparison (before and after the intervention in each network and between the intervention and control networks). To identify any changes after the intervention, data will be collected by: (A) a pre–post survey to participant doctors, (B) the patient survey and medical records audit (if available), using the methods and instruments of the baseline and (C) monitoring indicators. Impact on PC doctor's knowledge will be measured applying an anonymous on-line questionnaire survey before the training starts (baseline), at the end and 8 months after to measure any improvement in knowledge. Final outcome measures will be defined after the adaptation and testing of the intervention and may include intermediate healthcare outcomes (pathways in cancer diagnosis, time intervals, barriers of access) and final outcomes (provider delays and stage of cancer at diagnosis).

Two kinds of economic analysis will be carried out: (A) the implementation cost to estimate total costs of the implementation strategy and (B) a cost-effectiveness analysis of the intervention to calculate incremental costs per reduced diagnostic delay.

The evaluation will be completed combining the results of qualitative and quantitative studies, to identify the effectiveness of the intervention on early diagnosis of cancer and influencing factors in each country.

IV. Cross-country comparative analysis (30 months). Results will be compared across countries to identify contextual factors and actors that influence, on the one hand, access and equity of access to early diagnosis of cancer in different healthcare systems and on the other, the process and effectiveness results of the intervention; and, to inform the strategy to embed the intervention into local, regional and national health policy and practice.

To ensure these components develop according to plan and to an excellent level of quality and maximum societal impact, coordination and management activities, as well as activities for dissemination and exploitation of results and capacity building will run in parallel and will count on the involvement of the LSCs and the support of national and international scientific and international ethics advisory committee.

## Patient and public involvement

Patients are part, together with other key stakeholders, of the LSC and WG involved in the adaptation and implementation of the intervention, and specifically, of the component related to the patient information strategy to navigate the health services. They will also be involved in the dissemination of results.

## ETHICS AND DISSEMINATION
### Ethics approval

Comité Ético de Investigación Clínica (CEIC) del Parc de Salut Mar, Barcelona, Spain. Comité de Ética de Investigación con medicamentos (CEIm) de la Fundació Sant Joan de Déu, Barcelona, Spain. Comité de Ética de la Investigación de Salud Metropolitano Norte (CEI-SSMN), Santiago de Chile, Chile. Comité de Ética en Investigación de la Universidad del Rosario (CEI-UR), Bogotá, Colombia. Registration No.: 11 878 848 in Information, Research and Innovation Systems of the Universidad del Rosario. Comité de Ética de la Investigación en Seres Humanos CEISH-Pontificia Universidad Católica del Ecuador (PUCE) de la PUCE, Quito, Ecuador. Registration No.: 030-UIO-2021 in Research Register of the PUCE.

## Ethical issues

The development and execution of the project fully complies with all current international conventions and declarations,[83] EU legislation,[84 85] national legislation[86–98] ethical regulations, data protection and the professional code of conduct of each participating countries. Conditions of study procedure, risk and benefit evaluation, confidence and privacy, and informed consent were approved by the ethical committees in the participating countries. In addition, confidentiality agreements were signed with all participating institutions. Informed consent is obtained from every interviewee, after being informed that participation is voluntary and that they are free to refuse to participate without any negative consequence. Data are coded and processed in such a way that the individual cannot be identified, and appropriately stored. The project and the data processing comply with the European Union Data Protection Legislation and national legislation.

## Dissemination

Translating health research findings into practice and policy and disseminating them to the greater public in order to increase the impact on health system performance is an important challenge for any research. To tackle it, a comprehensive and efficient approach is adopted to maximise the research impact during and beyond the project, which will include strategies for dissemination and exploitation of results, communication and capacity building, as a transversal component of the project. A key element in it is the participatory approach applied at different levels, including policy, management, healthcare practice and research. Participatory research promotes healthcare practice and policy changes by bringing together the needs of the community and/or health services, scientific evidence and decision-makers, and co-producing knowledge with stakeholders who have the influence and decision-making capacity to implement the knowledge generated through research.[99] EquityCancer-LA expected impact will be achieved, in addition to involving key social actors, through the wide dissemination of results using the mechanisms, methods and materials appropriate to different audiences, as contained in the plan of dissemination: (1) to ensure that results inform policy-making to improve the early diagnosis of cancer: (A) policy briefs, short reports and guidelines. Moreover, a framework and strategy for the large-scale implementation of the intervention applicable in diverse contexts will be produced, as well as e-tools to improve clinical practice on diagnosis of cancer; (B) workshops and webinars to present key findings and policy recommendations to local and national interest groups in the three Latin-American countries and (C) building networks of key contacts (academic, governmental, non-governmental, civil society, including users' organisations, etc) in the participating countries and other countries in LA and elsewhere, and with international agencies, such as PAHO/WHO, EC, IAEA,

IARC, LA Cancer Institute Network and coordinating with established networks such as the Global Alliance for Chronic Diseases network; (2) for dissemination among the academic communities, research papers in open access peer-reviewed national and international journals and other relevant publications, as well as participation in national and international conferences; (3) website, external channels and media to promote visibility and awareness of the project.

**Author affiliations**
[1]Health Policy and Health Services Research Group, Health Policy Research Unit, Consortium for Health Care and Social Services of Catalonia, Barcelona, Spain
[2]Centro de Investigación Biomédica en Red de Epidemiología y Salud Pública (CIBERESP), Parc Sanitari Sant Joan de Déu, Sant Boi de Llobregat, Spain
[3]Centro de Investigación Biomédica en Red de Epidemiología y Salud Pública (CIBERESP), Research and Development Unit, Institut de Recerca Sant Joan de Deu, Barcelona, Spain
[4]Escuela de Salud Pública Dr. Salvador Allende Gossens, Facultad de Medicina, Universidad de Chile, Santiago de Chile, Chile
[5]Escuela de Medicina y Ciencias de la Salud, Universidad del Rosario, Bogota, Colombia
[6]Public Health Institute, Pontificia Universidad Católica del Ecuador, Quito, Ecuador
[7]NOVA National School of Public Health, Public Health Research Centre, NOVA University of Lisbon & Comprehensive Health Research Center (CHRC), Lisboa, Portugal
[8]Department of Public Health, Faculty of Health and Medical Sciences, University of Copenhagen, Copenhagen, Denmark

**Twitter** For EquityCancer-LA equitycancerla@equitycancerla1

**Acknowledgements** The authors are most grateful to the Local Steering Committees, managers, doctors and other professionals of the healthcare networks and research fellows that are participating in the study and generously share their effort, time and opinions, thereby making it possible. We highly appreciate the contribution of Laura Esteve and Andrea Miranda to the study protocol. We thank the European Union's Horizon 2020 research and innovation programme (SC1-BHC-17-2020) for the funding.

**Collaborators** List of collaborators of the study who, together with the authors of the paper, formed part of the EquityCancer-LA project led by M-LV (mlvazquez@consorci.org): Spain: Aida Oliver, Verónica Espinel, Zahara Fernández (Consortium for Health Care and Social Services of Catalonia), Josep Maria Borràs (Catalonian Cancer Strategy, Department of Health, Government of Catalonia), Montserrat Gil-Girbau, Antoni Serrano-Blanco, Paula Arroyo-Uriarte (Institut de Recerca Sant Joan de Déu); Chile: María Luisa Garmendia, Ana María Oyarce, Camilo Bass, Isabel Guzmán, Andrea Alvarez, Paola González (Public Health School Dr. Salvador Allende Gossens, University of Chile), Isabel Abarca, Rodney Stock, Berta Cerda, (National Cancer Institute), Guillermo Hartwig, Cristopher Tabilo (North Metropolitan Health Service), Carmen Aravena, Gloria Stephens (South Metropolitan Health Service); Colombia: Virginia Garcia, Ana Maria Pedraza, Angela Pinzón, Carol Cardozo, María Camila Rangel, Pablo Cristancho (School of Medicine and Health Sciences, Universidad del Rosario); William Mantilla (La Cardio), Guillermo León, Mauricio O'Byrne, Juan Merchán (Hospital Universitario la Samaritana); Ecuador: Iván Dueñas, Hugo Pereira, Daniel Ruiz, Estefanía Rodríguez (Public Health Institute, Pontifical Catholic University of Ecuador).

**Contributors** M-LV, IV, PE and ASM-P conceived the concept, objectives and the study design; M-LV and IV were responsible for the development of the study protocol, which received the contributions of the other authors (MR-V, IA-L, ALT-C, AP, SD and SSJ). IV and M-LV wrote the first draft of the paper. All authors reviewed the draft versions, made contributions and approved the final version of the article. The authors alone are responsible for the content of this paper.

**Funding** This project has received funding from the European Union's Horizon 2020 research and innovation programme under grant agreement No 965226 on the call topic SC1-BHC-17-2020, Global Alliance for Chronic Diseases-Prevention and/or early diagnosis of cancer.

**Competing interests** None declared.

**ORCID iDs**
Maria-Luisa Vázquez http://orcid.org/0000-0002-6091-8193
Ingrid Vargas http://orcid.org/0000-0002-1778-2411
Andrés Peralta http://orcid.org/0000-0002-7617-108X

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
