## [Reviewer comments · BMJ Open]

ARTICLE DETAILS

TITLE (PROVISIONAL)	Improving equity in access to early diagnosis of cancer in different healthcare systems of Latin America: protocol for the EquityCancer-LA implementation-effectiveness hybrid study
AUTHORS	Vazquez, Maria-Luisa; Vargas, Ingrid; Rubio-Valera, Maria; Aznar-Lou, Ignacio; Eguiguren, Pamela; Mogollón-Pérez, Amparo S; Torres-Castillo, Ana Lucía; Peralta, Andrés; Dias, Sónia; Jervelund, Signe

VERSION 1 – REVIEW

REVIEWER	Robb, Katie University of Glasgow
REVIEW RETURNED	13-Oct-2022

GENERAL COMMENTS	The protocol is very well written and describes an ambitious programme of work to improve equity in access to early diagnosis of cancer in Latin America. The rationale is clear with appropriate reference to relevant frameworks and guidelines. The authors may wish to additionally consider the following framework: Skivington K, Matthews L, Simpson S A, Craig P, Baird J, Blazeby J M et al. A new framework for developing and evaluating complex interventions: update of Medical Research Council guidance BMJ 2021; 374 :n2061 doi:10.1136/bmj.n2061 The methods seem appropriate although lacking in specific details possibly due to the scale of the work and limited word count. The research has been funded by the European Union's Horizon 2020 and has presumably been externally peer reviewed. Two minor suggestions: In the abstract methods, give Primary Care (PC) in full the first time it is used. In the Introduction, p. 9 "Understanding the extent to which an effective intervention for early diagnosis of cancer is so also in its specific health systems and social contexts, is challenging but needed and calls for implementation research." Please rephrase to improve clarity.
--

REVIEWER	Plackett, Ruth UCL
REVIEW RETURNED	17-Oct-2022

GENERAL COMMENTS	This is a well written and interesting protocol and I look forward to reading the final evaluation. I think the authors have identified an important research and practice gap and need for more evaluations of the implementation of an integrated care intervention to improve early diagnosis of cancer LMIC. There are
--

	some minor comments that I would encourage the authors to address.  1. In the abstract, it would be useful if the four phases were not split across sentences to make them clearer. Perhaps keep as one sentence and then add sentence on how participatory action sentence after. 2. In the abstract, perhaps the authors could include some examples of how the results will be disseminated instead of just saying a variety of ways. 3. In section need for implementation. I think there is a missing word between “so... also” 4. In the aim of the project section, I was not sure about the term contextual effectiveness as have not heard the term before, perhaps just delete those words in that sentence or describe what that means, as the term is used again? 5. In the aim of the project section, this is the first mention of equity-LA I and II, it would be useful if these were described (or the main findings) in the paragraph above on PAR perhaps? 6. In the study design section, should it say mixed-methods not mix-methods, more common term? Again may need to change this term in research phases section. 7. In the study design section, need to spell out what SIC acronym is. 8. In the study design section, could figure 1 be explained in more detail i.e. who are the actors, why these outcomes are defined as intermediate and final outcomes, could this be explained more clearly? 9. In study context section, which are the intervention and control areas for Columbia? 10. In research phases section, in phase 1 will focus groups be a mixture of different stakeholders in each focus group or 1 focus group per stakeholder group, how many focus groups or interviews anticipated, how many people in each? Which stakeholders will likely be interviewed which will likely be focus groups and why? How long will the survey be active for in phase 1? How will/were all the stakeholders recruited for the qualitative and quantitative parts in phase 1? 11. For phases 2-4 could some timelines be provided of when this is anticipated to be done. How will people be recruited for the LSC's?
--	---

VERSION 1 – AUTHOR RESPONSE

Reviewer: 1

Prof. Katie Robb, University of Glasgow

Comments to the Author:

The protocol is very well written and describes an ambitious programme of work to improve equity in access to early diagnosis of cancer in Latin America. The rationale is clear with appropriate reference to relevant frameworks and guidelines. The authors may wish to additionally consider the following framework: Skivington K, Matthews L, Simpson S A, Craig P, Baird J, Blazeby J M et al. A new framework for developing and evaluating complex interventions: update of Medical Research Council guidance BMJ 2021; 374 :n2061 doi:10.1136/bmj.n2061. The methods seem appropriate although lacking in specific details possibly due to the scale of the work and limited word count. The research

has been funded by the European Union's Horizon 2020 and has presumably been externally peer reviewed.

Thanks to the reviewer suggestion to consider the Skivington et al. framework(1), we realised that in the manuscript we failed to mention that in the design of the research phases, we had followed the Campbell model(2) for the evaluation of complex interventions. The reviewer's suggested framework(1) is an update of Campbell's original framework(2) and its further developments(3). After carefully reviewing the suggested article, we find that although Campbell's framework is clearer in defining the phases of the research, the updated framework highlights key elements for the participatory and implementation-effectiveness hybrid approach used by the EquityCancer-LA project. These include: the importance of the continuous evaluation of the implementation process and of the influence of the context to adjust the intervention, flexibility in adapting the design of the intervention during its implementation, and the involvement of key stakeholders in the tailoring, implementation and evaluation of the intervention. Therefore, we have decided to include the references of both frameworks(1,2) in the design of the research phases.

Two minor suggestions:

In the abstract methods, give Primary Care (PC) in full the first time it is used.

We have included Primary Care (PC) in full in the abstract.

In the Introduction, p. 9 "Understanding the extent to which an effective intervention for early diagnosis of cancer is so also in its specific health systems and social contexts, is challenging but needed and calls for implementation research." Please rephrase to improve clarity.

We have rewritten this sentence to clarify its meaning.

Reviewer: 2

Dr. Ruth Plackett, UCL

Comments to the Author:

This is a well written and interesting protocol and I look forward to reading the final evaluation. I think the authors have identified an important research and practice gap and need for more evaluations of the implementation of an integrated care intervention to improve early diagnosis of cancer LMIC. There are some minor comments that I would encourage the authors to address.

1. In the abstract, it would be useful if the four phases were not split across sentences to make them clearer. Perhaps keep as one sentence and then add sentence on how participatory action sentence after.

First of all, we would like to thank the reviewer for their comments and the suggestions received, which we have incorporated into the revised manuscript. Following the reviewer's recommendation, we have more rewritten the description of the project phases in the abstract.

2. In the abstract, perhaps the authors could include some examples of how the results will be disseminated instead of just saying a variety of ways.

Following the reviewer's recommendations and taking into account the word limit in the abstract, we have introduced some examples of the most important strategies for disseminating the results. These are described extensively in the dissemination section of the manuscript.

3. In section need for implementation. I think there is a missing word between "so... also"

We have rewritten this phrase to clarify its meaning.

4. In the aim of the project section, I was not sure about the term contextual effectiveness as have not heard the term before, perhaps just delete those words in that sentence or describe what that means, as the term is used again?

The term "contextual effectiveness" was taken from the H2020 call, where it was used both to describe the objective and expected impact of the research proposal. It refers, on the one hand, to the

evaluation of the effectiveness of the intervention under real-life conditions, and on the other hand, to the requirement that the interventions meet contextual factors identified as potential barriers. As this is not a commonly used term, we have clarified its meaning in the study design section and replaced it with “effectiveness under real-life conditions” elsewhere in the manuscript.

5. In the aim of the project section, this is the first mention of equity-LA I and II, it would be useful if these were described (or the main findings) in the paragraph above on PAR perhaps?

Equity-LA II implementation research aimed at evaluating the effectiveness of interventions – designed and implemented through participatory-action-research (PAR) processes – in improving clinical coordination and continuity between care levels in public healthcare networks of Argentina, Brazil, Chile, Colombia, Mexico and Uruguay(4). Equity-LA II built upon the theoretical framework and the results of Equity-LA, which explored access to the care continuum in healthcare networks of Colombia and Brazil.

Following the reviewer’s recommendations, in the introduction we have briefly described some of the Equity-LA II findings on contextual factors that may influence the implementation of integrated care interventions, using a participatory approach. And, to be more precise, in the objectives’ section, we have removed the reference to the Equity-LA as although Equity-LA II and EquityCancer-LA build on the Equity-LA frameworks, tools and results related to access an care coordination, those do not refer specifically to implementation research.

6. In the study design section, should it say mixed-methods not mix-methods, more common term? Again may need to change this term in research phases section.

We have replaced the term mix-method with the term mixed-method throughout the manuscript.

7. In the study design section, need to spell out what SIC acronym is.

The acronym SIC stands for: *Stages of Implementation Completion* (SIC) framework. We have introduced the full framework name in the manuscript.

8. In the study design section, could figure 1 be explained in more detail i.e. who are the actors, why these outcomes are defined as intermediate and final outcomes, could this be explained more clearly?

EquityCancer-LA analytical framework includes a) the evaluation of the intervention effectiveness and cost-effectiveness based on intermediate and final outcomes related to cancer diagnosis; and b) the analysis of the interventions implementation process and how it affects its effectiveness. The effectiveness analysis establishes any significant changes in the outcomes that can be attributed to the integrated care intervention to improve early diagnosis of cancer. Whereas the final outcomes have been defined according to the final objectives of the multicomponent intervention – mainly the reduction of diagnostic delay –, the intermediate outcomes relate to the proposed means to achieve the final outcomes – improvement of diagnostic skills in PC (recognition of symptoms and signs) and care coordination across care levels, and the establishment of clear and agreed fast-track referral pathways, complemented by a patient information strategy to navigate the system.

The analysis of both contextual factors and the implementation process, as well as the intervention results and costs will be carried out taking into account key *actors* (health professionals, managers and users) viewpoint.

We have provided now more details on the EquityCancer-LA conceptual framework for the evaluation of the intervention to improve early diagnosis of cancer

9. In study context section, which are the intervention and control areas for Columbia?

In Colombia, the study area is the department of Cundinamarca, located in the Andean region of the country. This department is made up of 116 municipalities, organised into 14 health regions. Two networks were selected for the study: the intervention network (IN) encompasses 8 health regions in the centre and north of the department, and the control network (CN), 6 health regions in the south. This information was summarized in the description of the study context in the manuscript.

10. In research phases section, in phase 1 will focus groups be a mixture of different stakeholders in each focus group or 1 focus group per stakeholder group, how many focus groups or interviews anticipated, how many people in each? Which stakeholders will likely be interviewed which will likely

be focus groups and why? How long will the survey be active for in phase 1? How will/were all the stakeholders recruited for the qualitative and quantitative parts in phase 1?

Thanks to the reviewer's comment, we realised that the description of the qualitative data collection in the manuscript was not accurate with missing information. Data collection was planned through semi-structured individual interviews with stakeholders (health professionals, administrative personnel, managers, policy makers, and cancer patients), and focus groups with some of these groups separately to complete/deepen the information gathered in the interviews, if considered necessary and feasible in the pandemic context. In the case of cancer patients, only individual interviews were considered, because this technique allows for a more adequate reconstruction of the individual patient's care pathway and further explore access barriers and facilitators based on their experience. We had planned to reach the sample size by saturation of discourses. The initial estimation was around 44 individual interviews (22 per network) and 4 focus groups (2 per network,) per country, at the time of writing the manuscript, the number of interviews conducted was more than double.

With regard to the duration of the patient survey, it was planned to run for approximately 5 months. For the selection of professionals, managers and policy makers, an institutional contact provided a list of possible candidates according to the selection criteria. Once selected, a first contact with them is made either directly or through the institutional contact to present the project and obtain their written informed consent to participate in the study. Patients are selected from the databases of patients diagnosed with cancer of the health services (in Colombia also from the insurers). Once selected, they are contacted and invited to participate by their health providers. Approval for participation and written informed consent is requested from informants prior to its participation. As described in the manuscript, in the case of the survey, patients are also selected from the health services' databases of patients diagnosed with cancer.

We have revised the description of the qualitative baseline study to provide more information on the sample, data collection techniques and selection of informants.

11. For phases 2-4 could some timelines be provided of when this is anticipated to be done. How will people be recruited for the LSC's?

Following the reviewer's recommendation, we have provided the duration of each phase of the research in the manuscript.

The Local Steering Committee (LSC) members represent all stakeholders involved in early diagnosis of cancer in the intervention network of each country. It consists of health professionals, managers and users of the intervention network, local policy makers and researchers. The LSC was set up at the beginning of the project and will remain active the five years of the project and will participate in all phases, being in charge of the adaptation and implementation of the intervention. It is key to ensure the institutional support and institutionalization of the interventions. The participating institutions designate their representatives to the LSC.

We have also included more information on the LSC, its members, functions and recruitment in the manuscript.

VERSION 2 – REVIEW

REVIEWER	Plackett, Ruth UCL
REVIEW RETURNED	24-Nov-2022
GENERAL COMMENTS	I would like to thank the authors for addressing our comments so thoroughly and look forward to the final evaluation paper.